# Lifestyle and Pharmacological Interventions and Treatment Indications for the Management of Obesity in Children and Adolescents

**DOI:** 10.3390/children10071230

**Published:** 2023-07-17

**Authors:** Despina Herouvi, George Paltoglou, Alexandra Soldatou, Christina Kalpia, Spyridon Karanasios, Kyriaki Karavanaki

**Affiliations:** Diabetes and Metabolism Clinic, 2nd Department of Pediatrics, University of Athens, “P&A Kyriakou” Children’s Hospital, 11527 Athens, Greece; d.herouvi@gmail.com (D.H.); alex_soldatou@hotmail.com (A.S.); xrkalpia@gmail.com (C.K.); spyroskara97@yahoo.gr (S.K.); kkarav@yahoo.gr (K.K.)

**Keywords:** children, adolescents, childhood obesity, treatment, lifestyle, pharmacotherapy

## Abstract

Obesity is a multifactorial chronic impairment that further decreases quality of life and life expectancy. Worldwide, childhood obesity has become a pandemic health issue causing several comorbidities that frequently present already in childhood, including cardiovascular (hypertension, dyslipidemia), metabolic (Type 2 diabetes mellitus, fatty liver disease, metabolic syndrome), respiratory, gastrointestinal and musculoskeletal disorders. In addition, obese children frequently experience stress and psychosocial symptoms, including mood disorders, anxiety, prejudice and low self-esteem. Given that cardiovascular risk factors and pediatric obesity have the tendency to pertain into adulthood, obesity management, including weight control and physical activity, should start before the late teens and certainly before the first signs of atherosclerosis can be detected. This review aims to concisely present options for childhood obesity management, including lifestyle modification strategies and pharmacological treatment, as well as the respective treatment indications for the general practitioner.

## 1. Introduction 

During the last decades, the prevalence and the severity of childhood obesity has been rising dramatically worldwide. Presently, according to epidemiological data, the prevalence of overweight/obesity in the age group under 5 years is 40 million children, while the relative prevalence for the age group between 5 and 19 years is over 340 million children [1]. Overall, it has been estimated that in the 5–19 age group the prevalence of obesity has increased from 11 million in 1975 to 124 million in 2016 [2]. However, there are reports indicating that the prevalence of childhood obesity in Western Europe, the USA, Japan and Australia has stabilized or even decreased, especially amongst girls and preschool children [3]. Nevertheless, the long-term tendency suggests that obesity in children of all ages will continue to increase [4].

Excess adiposity validly correlates with significant adverse health consequences on multiple organs/systems, including metabolic, endocrine, cardiovascular, gastrointestinal, pulmonary, neurological, psychological and skeletal complications [5].

Many studies have convincingly shown that childhood obesity is correlated with increased risk factors for cardiovascular (CV) diseases in adulthood, including hypertension, Type 2 diabetes (T_2_D), dyslipidemia, metabolic syndrome (MS) and early development of atherosclerosis [6,7,8,9]. Most alarming is that obesity itself and these risk factors are fairly stable in adolescents and tend to track into adulthood, unless the body mass index (BMI) improves [10]. In addition, excess weight in childhood has been found as an independent risk factor for various malignancies, such as colon, rectal and esophageal adenocarcinoma; endometrial and postmenopausal breast cancer; and sudden death [11,12]. The estimated decrease in life expectancy due to obesity is 0.8 to 7 years [13]. Moreover, the duration and severity of obesity in the pediatric population correlates positively with the risk for comorbidities and mortality in adulthood [14].

Emotional disorders related to childhood obesity include mood disorders, low self-esteem, anxiety and social isolation [15,16]. Children with excess weight are more likely to have fewer friends, lower school performance, more psychosocial problems and to be affected by bullying behaviors and social exclusion compared with their peers with normal weight [17]. In addition, unhealthy behaviors, such as alcohol or tobacco use, risky sexual behaviors, unfavorable dieting habits and physical inactivity are encountered more frequently among adolescents with excess weight [16].

Although the literature data show that lifestyle modification programs are effective only if applied intensively and continuously, they remain the cornerstone of obesity management. Pharmacotherapy has been recommended only for obese youth with serious comorbidities resistant to lifestyle changes [18]. Finally, bariatric surgery is the last alternative for adolescents with severe obesity, complicated by serious health problems, who have not been successful with lifestyle modification and medication [19].

In view of recent pharmacological treatment developments and new data regarding combined lifestyle and pharmacotherapy programs, we sought to create a concise review of non-surgical childhood obesity management options for the general practitioner. Therefore, the objective of this narrative review was to assess the efficacy of pharmacological interventions in the pediatric population, while lifestyle alteration programs were also discussed.

## 2. Methodology

This paper is based on a literature search in Pub Med for articles referring to the results of different strategies for the management of obesity in both children and adolescents, published between January 2002 and December 2022. Childhood was defined as age less than 18 years. The following keywords were used: children, adolescents, childhood obesity, treatment, lifestyle, pharmacotherapy. Three reviewers (C.K, S.K, G.P) performed the electronic search. Whenever the three reviewers disagreed with one another on the inclusion of an article, a fourth reviewer was recruited to decide on inclusion (A.S).

The initial selection of articles based on the relevance of the title yielded a total of 268 articles. From these, 0 were duplicates; thus, 269 records were screened. Among them, 150 records were excluded after reading the title or abstract for obvious irrelevance (as they referred to Type 2 diabetes, hyperlipidemia, hypertension, obesity epidemiology) and 119 were finally included for further full-text evaluation. Among them, 2 reports were not retrieved and, thus, 117 were assessed for eligibility. From them, 5 reports were excluded due to small sample size and 4 records because free full-text files were not available. Finally, 108 studies were included in the present review.

## 3. Prevention

Given the difficulty of losing weight through lifestyle intervention, as well as the subsequent weight maintenance and the potential harm of medication and surgery, obesity prevention should be a public health issue. It is therefore understood that childhood is an important period for preventing the development of obesity before it begins to create serious and irreversible health problems. In general, agreements for multi-sectional collaboration from governments, the private sector (food industries, advertising organizations, the media), civil society (consumer associations, public-interest organizations, academic institutions, research foundations, professional associations) and the public is mandatory in order to prevent obesity, as no single sector is able to solve the obesity burden on their own. Public health can have a primary role in coordination among local and national units, determining the principles that will establish a framework for effective obesity prevention strategies. These include individual behavior modifications; setting changes in multiple environments like homes, schools and workplaces; and sector change within food industries, advertising industries, agriculture, education, urban planning and transportation, such as:-The development of an urban planning policy, including access to green spaces, public transportation and school facilities;-Mass media campaigns and mobile apps to raise public consciousness of healthy eating and activity;-Reduction in junk food and beverage marketing, especially to children;-School-based programs combining nutrition and physical education, skill-building and policy changes;-Removing unhealthy foods from school canteens and vending machines (for example, since 2005 France has banned all vending machines from schools);-The provision of community-based strategies to inform people on healthy eating and physical activity, as well as support exclusive breastfeeding;-Regularly measuring BMI, counseling on healthy habits and prescription of physical activity by primary care doctors;-Action to alter the cost of healthy choices, such as taxes on unhealthy food products (such as sugary, energy-dense food and drinks) and subsidies for healthy choices (fruits, vegetables);-The provision of informed choices to people by labeling food products and restaurant menus;-Food reformulation and the development of healthier products, as occurred in 2018 in England, when Public Health England required the industry to reduce calories by 20% in foods high in sugar, salt and saturated fat by the year 2024 [20,21].

Considering that children aged 6–18 years spend a large part of their day in school, it is clear that the school environment is the ideal setting for childhood obesity intervention programs. The WHO school policy framework suggests the formation of wellness councils (by various school stakeholders), a published policy and programs supporting the adoption of healthy diets and physical activity. Schools should provide students with daily physical education and should have the necessary facilities and equipment. Governments should adopt policies that support the availability of healthy foods in schools and limit the availability of unhealthy products with high salt, sugar and fat content [22]. 

Nowadays, in many countries, including the UK, the USA and Australia, there has been significant progress in preventing childhood obesity by setting prevalence goals, establishing national guidelines, streamlining surveillance systems and promoting public education (through community events, mass media and social media campaigns) [23]. The participation of teachers, trained by health professionals, in promoting healthy behaviors and acting as role models; the provision of additional hours for physical activity (e.g., an extra 30 min of physical activity/school day); the creation of after-school physical activity programs and education classes that support healthy eating habits; parental engagement in nutrition education and food preparation skills; activities such as dance classes and cooking lessons; and the mobilization of stakeholders to find the required funds for healthy breakfast, lunch and snack programs in schools have been suggested as effective strategies [22,23].

## 4. Treatment

Obesity is a long-lasting, relapsing disorder that necessitates longitudinal care provided by an interdisciplinary group of trained health-care professionals, including pediatricians, dieticians, psychologists, exercise instructors and social workers. The participation of families, schools, communities and health policy makers is required to ensure coordination and universal accessibility. Guidelines for the treatment of pediatric overweight/obesity take into consideration various factors, such as the patient’s age, the degree of adiposity and the presence of comorbidities. The management of obesity requires behavioral changes, diet modifications, improved quality of sleep and increased physical as well as decreased sedentary activities (Figure 1). Intervention programs often include family participation, especially for children under the age of 12 years, given that parents influence children’s eating and physical activity habits significantly [24].

The most effective behavioral approaches are goal setting, positive reinforcement, self-monitoring, contingency training, stimulus control, cognitive restructuring, behavior chains, problems solving and relapse prevention. Treatments such as appetite-awareness training and the regulation of cues are considered experimental. According to the US Preventive Services Task Force, a multicomponent intervention program is usually most effective (with a reduction in the BMI z score ≥ 0.20) when it occurs in person, engages the entire family and delivers at least 26 h per year of nutrition, physical activity and behavior-change sessions over 3 to 12 months [25]. However, it is difficult to implement such interventions since they are costly and require a multidisciplinary team with specific training.

Nevertheless, in severe cases of obesity, pharmacotherapy, and even bariatric surgery, may be considered as alternative treatment options [18]. Bariatric surgery is a viable option for adolescents with severe obesity and relevant medical comorbidities who have not seen weight loss with conventional lifestyle and pharmacologic interventions. However, since the majority of children with high BMI do not exhibit life-threatening comorbidities, the risk–benefit ratio of metabolic surgery is difficult to assess [19]. 

### 4.1. Dietary Management

The mainstay of obesity treatment is dietary management. Generally, typical weight-management programs are based on the reduction of caloric intake in order to induce a moderate weight loss of 0.5 kg per week [26]. A classic example of a diet aiming at controlling calories is the Traffic Light Diet, according to which foods are color-coded so as to separate those one might eat freely (green) from those one should consume more cautiously (amber and especially red) [27]. A large research body focuses on determining the most efficient distribution of macronutrients for successful weight loss, along with other potential metabolic benefits, with conflicting results.

Low-carbohydrate (10–30% of caloric intake from carbs) and low-fat (18–40% caloric intake from fat) diets have been proven efficacious in the short-term [28]. These diets presumably foster satiety, resulting in lower caloric intake. A 12-week trial showed that high-protein, low-carbohydrate (HPLC) diets had greater short-term efficacy than low-fat (LF) diets. However, by the end of the study, the efficacy of both diets was similar [29]. Furthermore, markers of cardiovascular risk and insulin resistance improved in both groups, with the HPLC diet demonstrating greater results of insulin resistance markers. The potential superiority of low-glycemic-index diets compared with energy-restricted, low-fat diets in body-weight reduction has been shown in several pediatric studies, probably due to delayed insulin secretion, which may in turn cause prolonged satiety. The DiOGenes randomized controlled trial demonstrated protective effects of the combined use of a high-protein (HP)/low-glycemic-index (L-GI) diet against obesity [30].

Very low-calorie diets or VLCDs, i.e., hypo-caloric diets that provide 800 calories or less per day, are only used with severe forms of obesity. Since VLCDs aim to achieve significant weight loss without severe electrolyte imbalance, medical supervision is recommended when the duration of treatment lasts over 3 weeks [31]. Due to expeditious results in weight loss, improvements in body composition and metabolic parameters, VLCDs could be considered as alternatives to pharmacological and/or surgical interventions for adolescents with severe obesity. Therefore, the long-term safety and comparable beneficial effects of these diets in pediatric populations merit further research.

Prebiotic supplementation has shown promise in managing pediatric obesity with notable improvements in satiety and lower energy intake [32]. Diet is a well-established effective short-term intervention for achieving weight loss and improvement of the cardio metabolic profile [33,34]. Comparison of the outcome of a hypo-caloric diet vs. exercise showed improvements in BMI, blood lipids and adiponectin levels in both groups at 12 weeks and greater reductions in BMI and LDL levels in the diet-only group. Notably, both groups regained weight 9 months after the intervention [33].

Evidence suggests energy-restricted diets improve the weight status of children and adolescents, regardless of the macronutrient proportions [35,36]. Thus, focusing on the attainment of sustained adherence to a balance and varied diet is more important than the type of macronutrients’ alteration for successful weight loss. In order to help children embrace recommended eating patterns, we could involve them in meal planning, shopping, gardening and the preparation of food. However, diet modification alone is not sufficient to achieve weight loss. When caloric intake decreases, metabolism slows, resulting in decreased calorie utilization and difficulty achieving weight loss. That is why physical activity is vital for successful weight loss [37].

### 4.2. Physical Activity

There is robust evidence regarding the benefits of regular physical activity in childhood and adolescence with a wide range of effects, such as motor skills’ development, healthy weight and body composition enhancement, and bone and muscular development. Moreover, physical activity improves cardio respiratory capacity, cardiovascular and metabolic health biomarkers, immunity and quality of life [38]. Thus, it is a main component of all weight-management programs, particularly in the phase of weight maintenance. Weight loss induced by diet escalates appetite and energy intake by increasing ghrelin and reducing insulin and leptin levels, whereas exercise adapts the metabolism and improves energy balance regulation [39]. When children and adolescents with obesity participated in a 30 to 60 min physical activity program for 3 to 7 days a week, total body fat and visceral adiposity were reduced; the same program had no effect on the body fat of normal-weight participants [40]. The results of further studies are inconclusive; some studies report a significant decrease [41] and others report no change in adiposity [42].

Although physical fitness’s (PF) role in weight loss remains uncertain, its benefits for overall health are unequivocal. Thus, the existing literature indicates that increasing the level of PF dampens the metabolic consequences associated with excess adiposity and has positive effects on mental and skeletal health [43]. It is well-established that exercise promotes favorable changes in body composition and cardiovascular parameters, such as blood pressure, serum glucose, insulin levels and lipid profile [34,44,45,46], as seen in Table 1. Moreover, physical activity dampens the obesity-related inflammatory state, promoting an increase in adiponectin levels and a decrease in both C-reactive protein (CRP) and Interleukin-6 (IL-6) levels [7], regardless of the associated weight loss or body composition changes [47]. Finally, even without weight loss, exercise improves or even normalizes flow-mediated dilation (FMD) and intima media thickness (IMT). Both abnormal FMD and IMT are well-documented indicators of the initiation of atherosclerosis, reflecting arterial function and vascular wall structure, respectively [48]. A physically active lifestyle appears to halt subclinical atherosclerotic vascular changes in healthy adolescents and can improve arterial function and structure in children with cardiovascular risk factors [49]. Although aerobic and resistance training similarly decrease body fat, only resistance exercise accelerates insulin sensitivity significantly [50]. Notably, combined aerobic and resistance training shows superior effectiveness than either alone in improving various adiposity measures (e.g., BMI, total body fat), fasting glucose levels, metabolic profile and adiponectinemia [51]. Additionally, in a group of adolescent females, high-intensity exercise had a greater positive effect on blood lipids, adiponectin levels and insulin sensitivity than moderate-intensity exercise [52]. Thus, although the beneficial effects of exercise are indisputable, the ideal weight-loss exercise program has not been established so far.

### 4.3. Diet Plus Exercise

Since the implementation of single-treatment options to improve weight status and BMI has limited results, exercise combined with diet and behavioral adaptations is regarded by most experts as optimal for weight loss [58]. Dietary restriction along with physical exercise act synergistically to decrease body mass while maintaining an unchanged resting energy expenditure [53,54,55,56,57,59,60,61]. For example, following a 12–24 month intervention, a decrease in BMI z-score ranging from 0.05 to 0.42 was observed [60]. On the contrary, a review of randomized controlled trials suggested that combined lifestyle modification interventions may lead to small, short-term decreases in BMI and weight in school-aged children [61]. However, maintenance of the same weight with linear growth in children means a reduction in BMI, as well as an improvement in cardiovascular risk factors [62]. Lifestyle modifications have particularly limited longer-term results among adolescents with severe obesity [63].

A structured, comprehensive, multidisciplinary and personalized lifestyle intervention program of healthy diet and physical exercise for one year resulted in a significant increase in leukocyte telomere length in children and adolescents, irrespective of gender, pubertal status or BMI, as well as a significant improvement in metabolic syndrome parameters, including clinical and body composition indices of obesity, inflammatory markers, hepatic enzymes, markers of insulin resistance and lipid profile [64].

A recent meta-analysis of lifestyle interventions outcomes confirmed the positive effects of diet and diet plus exercise on weight, blood pressure and metabolic parameters such as LDL cholesterol, triglycerides and fasting insulin for up to one year [65]. When diet and diet plus exercise were compared, a greater reduction in triglycerides and LDL levels with diet and a greater improvement in HDL, fasting glucose and insulin levels with diet plus exercise were shown. Favorable effects of diet plus exercise or exercise alone on endothelial function and cardiovascular risk factors, including blood pressure, lipid profile, insulin resistance, metabolic syndrome and inflammatory status, have also been reported (Table 1).

However, the efficacy of lifestyle interventions appears to depend on the age of the patient at initiation and the severity of obesity. Specifically, although behavioral treatment was effective for children with moderate and severe obesity, it had no significant effect for adolescents with severe obesity [66]. A recent study highlighted young age at treatment onset as the most important predictive factor for weight reduction, underlining the value of early interventions as the cornerstone for children’s natural growth and weight-related habits [67]. Importantly, lifestyle interventions do not carry the risk of serious side-effects or health inequalities [68]. Therefore, obesity should be addressed at the earliest time point in life.

### 4.4. Pharmacological Treatment

The National Institute for Health and Care Excellence Guidelines 2014 state that medication is not generally recommended for children younger than 12 years, except for those with life-threatening co morbidities, such as intracranial hypertension and sleep apnea. Pharmacotherapy along with lifestyle modification is considered appropriate for adolescents over 12 years of age with obesity (BMI ≥ 95th percentile) and further weight gain in spite of a 12-month program of earnest lifestyle modifications and only when physical co morbidities (such as orthopedic problems or sleep apnea) or serious psychological problems co-exist [69]. It is worth emphasizing that researchers like Styne et al. suggested that pharmacotherapy should discontinue if the patient does not have at least a 4% BMI reduction after using anti-obesity medication for 12 weeks at full dosage. Additionally, medication should be administered exclusively at tertiary institutions where multi-disciplinary planning and support is provided to the patient [70]. Currently, orlistat and Glucagon-like peptide 1 receptor agonists are the only FDA-approved treatment against childhood obesity, despite the off-label use of other medications. The different pharmacological agents used for the management of childhood and adolescent obesity are shown in Table 2.

#### 4.4.1. Sibutramine and Orlistat

Sibutramine is a serotonin- and noradrenergic-reuptake inhibitor. Compared with placebo, it led to statistically significant weight loss, particularly if combined with lifestyle modification [18]. However, sibutramine has been removed from both US and Europe markets since 2010, due to its association with an elevated risk of adverse cardiovascular side effects, such as myocardial infarction and stroke [71,80].

Orlistat acts in both the stomach and intestine as a reversible gastric and pancreatic lipase inhibitor. It prevents dietary triglyceride absorption, thus creating a negative energy balance. Reportedly, fat absorption decreases by 30%. Currently, only orlistat meets the age-restricted approval of the FDA for long-term treatment of obesity at a dosage of 120 mg three times per day for children older than 12 years [81].

According to limited data, orlistat is well-tolerated, safe and results in greater weight loss than lifestyle modification alone. Specifically, except for one, several well-documented clinical trials on the use of orlistat in comparison with placebo including pediatric patients reported statistically significant decreases in BMI from baseline, ranging from 0.55 to 4.09 kg/m^2^. In addition, there were no important differences in the levels of fat-soluble vitamins between groups during the trials [82,83,84].

In addition to weight loss, obesity treatment should improve cardiovascular and metabolic risk factors. Thus, orlistat has been shown to contribute to the reduction of total cholesterol, LDL-cholesterol and fasting insulin levels. In addition, adolescents on orlistat had significantly improved diastolic blood pressure compared with those on placebo [82]. Furthermore, the combined use of orlistat and dietary measures resulted in the improvement of endothelial function, as well as the reduction of BMI and fasting total- and LDL-cholesterol amongst adolescents suffering from obesity [77].

The most commonly observed side effects of orlistat are gastrointestinal, including steatorrhea, abdominal pain, nausea, flatulence, fecal urgency, diarrhea and gallstone formation. Since the malabsorption of fat-soluble vitamins (A, D, E, K) and b-carotene is particularly risky for developmental growth as well as sexual maturation during adolescence, the FDA recommends concurrent daily multivitamin supplementation [85].

The duration of randomized, controlled trials examining the use of orlistat is up to 12 months of therapy. Therefore, longer-term assessments of efficacy and safety are necessary. Co-administration of orlistat with other medications is not indicated, while the obligate increase in undigested stool triglycerides may cause considerable adverse gastrointestinal effects, driving many patients to discontinue treatment [86].

#### 4.4.2. Metformin

Metformin is an effective orally administered hypoglycemic and insulin-sensitizing agent for the treatment of T_2_D in both adults and children above 10 years of age [87]. Moreover, metformin inhibits fat-cell lipogenesis and decreases food intake by increasing the levels of glucagon-like Peptide 1 (GLP-1), leading to satiety, followed by weight loss [88]. Despite the lack of approval for the treatment of obesity in children and adults, metformin may also be helpful in addressing other insulin-resistant states, including fatty liver disease, polycystic ovary syndrome and MS [89,90,91], as well as weight gain induced by the use of psychotropic medications in pediatric patients [92].

According to accumulating data, metformin reduces the progression from impaired glucose tolerance to T_2_D amongst non-diabetic hyperinsulinemic pediatric patients [93]. In addition, metformin was found safe enough and well-tolerated, despite mild and transient gastrointestinal side effects, most commonly metallic taste, mild anorexia, nausea, abdominal discomfort and diarrhea. Associated vitamin B12 deficiency is managed with the concurrent intake of a multivitamin compound. Lactic acidosis is presumably an uncommon side effect that is mainly observed in adult patients [94,95].

In a study by Kendall et al. on subjects with high BMI, hyperinsulinemia and/or increased fasting glucose or impaired glucose tolerance, participants were randomized to receive either metformin or placebo, in addition to diet and exercise counseling, for 6 months. The metformin group achieved decreases in BMI and fasting glucose levels in three months, with maintenance of only the decreased BMI six months into the trial [94]. A longer-term study of obese adolescents reported a statistically considerable decline in BMI in the metformin group but no effect on insulin resistance (IR) after 48 weeks of follow-up. Notably, weight loss lasted 12 to 24 weeks, even after discontinuing pharmacotherapy [95]. To date, the largest trial subtended 955 children and adolescents with obesity and evidence of IR, divided into three groups: routine counseling alone, intensive counseling and metformin plus intensive lifestyle interventions. At 6 and 12 months, the metformin group experienced significant reductions in BMI, which continued up to 24 months [96]. Furthermore, metformin may be associated with positive secondary outcomes including reduced inflammatory and metabolic disturbances, such as the adiponectin–leptin ratio, interferon-γ and total plasminogen activator inhibitor-1, CRP, IMT and liver fat [72,87].

#### 4.4.3. Glucagon-like Peptide 1 Receptor Agonists

Although glucagon-like peptide 1 receptor agonists (GLP-1a), like liraglutide and exenatide, were originally used for the treatment of patients with T_2_D, newly published data confirmed their ability to induce weight loss. These agonists increase insulin secretion and inhibit glucagon release only when glucose levels are elevated, while they delay gastric emptying and suppress appetite. In addition, adult studies suggest protective effects of GLP-1a on multiple organs, with favorable changes in hypertension, myocardial contractility, insulin sensitivity, endothelium function, lipid profile and hepatosteatosis [97]. Liraglutide was FDA-approved in 2014 for the treatment of adults with obesity and in 2019 for the treatment of children and adolescents older than 10 years with T_2_D. Presently, liraglutide is the only GLP-1a that received FDA-approval in 2020 for use in obese children aged ≥12 years at a daily dose of 3 mg [78]. Due to the rising prevalence of obesity, the increased desire for weight loss treatments has contributed to the market’s expansion. Thus, the global Liraglutide injection market generated revenue of around USD 4.4 billion in 2021 and is anticipated to grow over 9.3% during the following period, 2022–2028, to reach USD 7.5 billion in 2028 [98].

Liraglutide therapy in adolescents with high adiposity was associated with a reduction in both weight and BMI and an improvement in metabolic profile with mild side effects [99]. A non-randomized, open-label study of liraglutide for a 36-week duration in an adolescent population with obesity concluded that approximately a quarter of patients had >10%, half had between 5% and 10% and another quarter had <5% weight loss [100]. Similarly, a recent randomized controlled trial showed that the combination of liraglutide plus lifestyle intervention promotes a significantly greater reduction in the BMI z-score compared with the placebo plus lifestyle modification program. Adverse effects included gastrointestinal symptoms and mild hypoglycemic episodes, with no effect on pubertal development or growth [101]. Presently, there is an ongoing trial testing the efficacy and safety of liraglutide as an anti-obesity drug in children aged 6–12 years, which is to be completed in January 2024 [102].

Moreover, 3–6 months of exenatide administration in adolescents with severe obesity induces a modest reduction in BMI and improvement in fasting insulin, insulin sensitivity and cholesterol [79,103]. In addition, the FDA has recently approved semaglutide to be used in weekly doses of 2.4 mg/week for adolescents with obesity aged ≥12 years [104]. Another medication in long, large Phase 3 clinical trials of efficacy for weight management is tirzepatide, a glucose-dependent insulinotropic polypeptide (GIP) and GLP1R dual-agonist that takes advantage of GIP-GLP1R synergism and also appears to have substantial beneficial effects on body weight and glucose homeostasis in patients with Type 2 diabetes with adverse consequences that generally did not lead to discontinuation [78].

#### 4.4.4. Other Anti-Obesity Medications

There are individualized medication options for children with obesity and underlying genetic and/or metabolic disorders. For example, children with hereditary leptin deficiency receive recombinant leptin [105] and those with Prader–Willi syndrome receive growth hormone [106]. Octreotide, a somatostatin agonist, diminishes ghrelin levels, thus inducing weight loss in children suffering from hypothalamic obesity [73]. Additionally, in 2020 the FDA approved setmelanotide, a selective agonist of the melanocortin-4-receptor (MC4R), for the treatment of monogenic obesity in subjects aged 6 years and above. Two years later, it was also approved for use in Bardet–Beidel syndrome [107].

Other FDA-approved medications for the treatment of adult obesity (e.g., phentermine, topiramate) have undergone investigation in the context of pediatric obesity. Phentermine, an anorexiant agent, is FD-approved for weight loss in individuals >16 years of age. A retrospective review of 299 adolescents with obesity receiving phentermine in addition to lifestyle interventions demonstrated significant decreases in BMI at 1-, 3- and 6 months [74]. Topiramate is FDA-approved for epileptic seizures in children ≥2 years of age and has been studied for the treatment of obesity in adults. Two clinical trials of topiramate in adolescents with severe obesity demonstrated decreased BMI (2–4.9%) for the topiramate group compared with placebo [75].

Since monotherapy has limited efficacy, combined pharmacotherapy merits further evaluation [108]. Therefore, larger studies with stricter methodological criteria, longer duration and patient follow-up are needed to evaluate the long-term safety and potential positive effects of combined pharmacotherapy in childhood and adolescent obesity [Table 2].

## 5. Conclusions

Obesity is a multi-factorial chronic impairment. Despite a recently observed stabilization regarding the prevalence of pediatric overweight and obesity in developed countries, overall levels remain unacceptably high. Childhood obesity is associated with serious medical co morbidities that probably persist into adulthood, whereas the nature of the link between early high BMI and adult disease risk is very complex.

Lifestyle alteration programs should always be considered as first-choice therapy. The combination of dietary restriction along with physical exercise has presumably synergistic results, leading to decreased body mass without a corresponding reduction in resting energy expenditure. The literature suggests that comprehensive obesity management programs promote modest weight loss that varies with the intensity and duration of the treatment.

Although still debatable, anti-obesity medication could reduce body weight, contributing to the remission of obesity-related co morbidities. Further long-term studies are necessary to determine the efficacy and safety of new anti-obesity medication for children and adolescents.

## Figures and Tables

**Figure 1 children-10-01230-f001:**
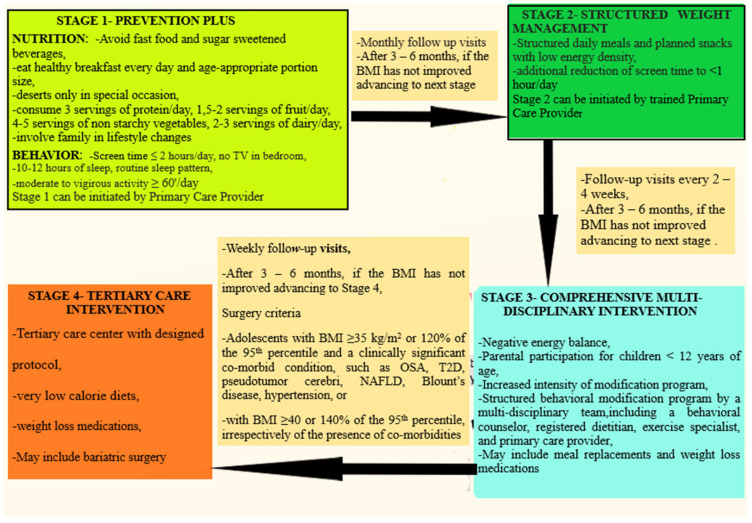
Clinical algorithm for childhood obesity treatment [19,24].

**Table 1 children-10-01230-t001:** Summary of intervention studies in obese/overweight children and adolescents.

Study	Population	Age (Years)	Intervention	Length of Intervention/Follow-Up	Outcomes
Zorba et al. [44]	40 obese children	10–12	Aerobic exercise	3 months/-	Improvements in BMI, LDL, HDL, VLDL, insulin levels
Park et al. [45]	29 overweight/obese children	12–13	Aerobic + resistance exercise	3 months/-	↓ BMI, ↓ SBP, ↓ IMT
Calcatera et al. [46]	22 obese children/adolescents	9–16	Aerobic + resistance + strength exercise	3 months/-	↓ BMI, ↓ WC, ↓ SBP, ↓ TG, ↓ HOMA-IR, ↓ Glu and improvement in CRF
Kelishadi et al. [53]	35 obese children/adolescents	12–18	Aerobic exercise + dietary restriction	6 weeks/-	↓ BMI, ↓ %BF, ↓ WC, ↓ CRP, ↓ FI, ↓ ox-LDL,↑ FMD
Ben qunis et al. [54]	28 obese children	13.2 ± 0.7	Exercise (running, jumping, playing with balloon) + dietary restriction	2 months/-	↓ BMI, ↓ %BF, ↓ CRP, ↓ IL-6, ↓ TNF-a, ↓ leptin, ↓ IGF-1, ↓ IGFBP-3
Luo et al. [55]	215 obese children	11–13	Aerobic exercise + dietary restriction	6 weeks/-	↓ BMI, ↓ WC, ↓ SBP and DBP, ↓ TG, ↓ HOMA-IR, ↓ FG (only girls), ↓ FI
Van der Baan et al. [56]	90 obese children/adolescents with comorbidities	8–18	Aerobic exercise + nutrition + behavior modification	6 months/2 years	↓ BMI z score, ↓ TC, ↓ LDL, ↓ TG, ↓ HOMA-IR, ↓ FI
Sigal et al. [57]	304 obese or overweight adolescents with CV risk factors	14–18	Dietary restriction + exercise (aerobic, anaerobic, combined), no exercise	6 months/-	↓ BMI, ↓ WC, ↓ SBP and DBP, ↓ TG, ↓ HOMA-IR, ↓ FG (only girls), ↓ FI

Abbreviations: %BF: percentage of body fat, WC: waist circumference, SBP: systolic blood pressure, DBP: diastolic blood pressure, ox-LDL: oxidized LDL, V_O2_ max: maximal oxygen uptake, CRF: cardio respiratory fitness, FG: fasting glucose, IGF-1: insulin-like growth factor-1, IGFBP-3: insulin-like growth factor-binding protein-3, FI: fasting insulin, HOMA-IR: homeostasis model assessment for insulin resistance.

**Table 2 children-10-01230-t002:** Pharmacological agents used for weight loss in children and adolescents, including those taken off the market.

Name	Medication Class	Mechanism of Action	Side Effects	Status
**Ι. Agents withdrawn or not FDA-approved for childhood obesity**				
Sibutramine [71]	B-phenethyl amine	Noradrenergic-serotonergic causing enhancement of satiety, stimulates thermogenesis	Insomnia, dry mouth, headache, constipation, increases blood pressure and heart rate, adverse cardiovascular effect (stroke and infarction) and cardiovascular death	Withdrawn in 2010
Metformin [72]	Biguanide antihyperglycemic	AMP-activated protein kinase activator, anorexiant	Abdominal discomfort, nausea, diarrhea, metallic taste, infections of upper respiratory tract.	FDA-approved for treatment of T_2_D and PCOS in subjects ≥10 years of age. Not FDA-approved for obesity
Octreotide [73]	Somatostatin analog	Appetite suppressant, inhibitor of insulin secretion, reduces ghrelin levels	Abdominal pain, diarrhea, gallstone formation, nausea, cardiac abnormalities, B12 deficiency, hypothyroidism, suppression of growth hormone secretion	Not FDA-approved for obesity, promotes weight loss in children with Prader–Willi syndrome and hypothalamic obesity
Phentermine [74]	Sympathomimetic	a norepinephrine reuptake inhibitor causing appetite suppression	Insomnia, dry mouth, taste alteration, dizziness, headache, anxiety, palpitations, hypertension, blurred vision, cardiac arrhythmias, diarrhea, constipation	Approved for short-term use in subjects ≥16 years of age
Topiramate [75]	Antiepileptic	GABA receptor activator/glutamate receptor inhibitor causing appetite suppression, reducing food intake or stimulating energy expenditure	Anorexia, fatigue, somnolence, paresthesia, psychomotor disturbances, difficulty with concentration/attention, metabolic acidosis, teratogenicity (↑ risk of oral palate in fetus)	Approved for epilepsy ^1^ and migraine prophylaxis ^2^ in children and adults. Not FDA-approved for obesity as monotherapy
Fenfluramine [76]	Antidepressant	Serotonergic causing appetite suppression	Insomnia, drowsiness, tremor, short term memory loss, valvular heart disease, primary pulmonary hypertension	Withdrawn in 1997
**ΙΙ. Agents FDA-approved for childhood obesity**				
Orlistat [77]	Gastro-intestinal lipase	Reduces the absorption of ~30% of ingested dietary fat	Abdominal pain, nausea, diarrhea, steatorrhea, oily stools, flatulence, gallstone formation and malabsorption of fat-soluble vitamins	FDA-approved for obesity in subjects ≥12 years old
Liraglutide [78]	An analog of human GLP-1	Increases insulin secretion/inhibits glucagon release. Delays gastric emptying and decreases food intake	Diarrhea, vomiting, nausea, constipation, headache, increases in BP and HR, pancreatitis, gallbladder disease, renal impairment, depression, suicidal behavior	FDA-approved for obesity in subjects ≥12 years old and for T_2_D in subjects ≥10 years of age
Exenatide [79]	An analog of human GLP-1	Increases insulin secretion/inhibits glucagon release. Delays gastric emptying and decreases food intake	Diarrhea, vomiting, nausea, constipation, headache, renal impairment, pancreatitis, gallbladder disease, teratogenicity	FDA-approved for treatment of T_2_D in adults and children ≥10 years of age

Abbreviations: AMP: adenosine monophosphate, PCOS: polycystic ovary syndrome, BP: blood pressure, HR: heart rate. ^1^ in adults and children 2 years and older. ^2^ in adults and children 12 years and older.

## Data Availability

Not applicable.

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
