# Peer review of "Lifestyle and Pharmacological Interventions and Treatment Indications for the Management of Obesity in Children and Adolescents"

_children, 2023, doi:10.3390/children10071230_

Round 1

Reviewer 1 Report

This manuscript proposed a review focuses on the lifestyle alteration –should be modifications-, pharmacological interventions and treatment indications for the management of obesity in childhood and adolescence.  This is a relevant topic, but some issues were faced during the reading of the manuscript. 

Initially, I would recommend the authors to reorganize the information and ideas presented in the different sections, as some of them are repetitive. In this sense, section 3 should be summarized and incorporated as part of the introduction. The information regarding the situation in Greece seems out of context, since the title suggests that the review is not limited to one region or space. 

One of the biggest concerns is regarding the methodology, despite this is a review but not a systematic one, the search strategies should be clarified and described, although they suggest that they will include information related to children and adolescents, the criteria do not suggest an appropriate search. Also, there is no information regarding the flow of the through the different phases of  the review. How many articles were identified, screened, elegible and  included. Results should start with the complete number or files included, as well as the rate of selection. 

I consider the main problema of the manuscript is the way information is presented, some visual resource (table or image) could be included to help identify the different possibilities of therapy or interventions (diet, physical activity, screen time, sleep quality and pharmacological therapies, individually or as a combination of strategies). 

There are classifications of interventions according this variables or lifestyles modifications; those school-based, home only-based, home-based with school components, community-based interventions or environmental-level, primary care only-based… to mention a few, as well as the level of intervention (individual, interpersonal, organizational, community or as public policy) or variables considered as part of the management of obesity among this population, and mentioned above (some of them).  

In this sense the results are not clear or exhaustive and did not present the proposal made as part of the “integral” treatment of childhood obesity.  I would recommend improving this section.  

On the other hand, I would highlight one of the main strengths of the pharmacological section, as it includes the alternatives used or discarded as a treatment, I just recommend to reorganize the table as those recommended or approved and those withdrawn.   

Some manuscripts you could consider are:  

Wang Y, Cai L, Wu Y, Wilson RF, Weston C, et.al. What childhood obesity prevention programmes work? A systematic review and meta-analysis. Obes Rev. 2015 Jul;16(7):547-65 

Writing Group for the National Collaborative on Childhood Obesity Research (NCCOR). Developing a Partnership for Change: The National Collaborative on Childhood Obesity Research. Am J Prev Med. 2018 Mar;54(3):465-474 

Gurnani M, Birken C, Hamilton J. Childhood Obesity: Causes, Consequences, and Management. Pediatr Clin North Am. 2015 Aug;62(4):821-40 

Finegood DT, Merth TD, Rutter H. Implications of the foresight obesity system map for solutions to childhood obesity. Obesity . 2010 Feb;18 Suppl 1:S13-6.  

Reviewer 2 Report

Dear Authors, 

This review on the lifestyle and pharmacological interventions for the management of obesity in children and adolescents describes an important topic. However, I believe this paper doesn´t qualify for acceptance at this time. The introductory section needs to be improved and clearly highlights the novelty of the paper. There are many reviews of its kinds (e.g., Ann Pharmacother. 2015 Feb;49(2):220-32; Nat Rev Endocrinol. 2013 Oct;9(10):607-14; Arch Dis Child Educ Pract Ed. 2013 Jun;98(3):108-12; Paediatr Child Health. 2020 Aug 20;26(5):310-316). How this review extends reader understanding of the topic. What is new? What this review adds? An introductory paragraph about obesity prevalence should be relatively short. A good paragraph should be brief and focused. It would be benefits to provide evidence about the risk factors for obesity/overweight in children and adolescents (e.g., unhealthy eating habits, sedentary behaviour…etc.). The paper lacks acknowledgement of extensive behaviour change theories and techniques used to inform lifestyle and pharmacological interventions for children. I miss a figure that helps to understand a visualize the key points. What policy action is needed? What players need to come together? English language should be improved throughout.

Reviewer 3 Report

The authors present a complete review of the up-to-date possible interventions to treat childhood and adolescence obesity. Their review is easy to follow, and of possible interest even for colleagues not espeially keen on pediatric obesity. My one and only remark (a minor one anyway) regards the marketing of the drug liraglutide, as stressed in a comment in the attached file.

Reviewer 4 Report

This systematic review focuses on the validity and the efficacy of combined intervention to prevent, cure, and reverse pediatric overweight/obesity. Its scope and outcomes are appealing and of interest, especially considering the pandemic scenario due to overweight & physical inactivity, and the subsequent counter-measures enacted.

My major comments are appended below:

1/ Introduction and background would benefit from citing other important studies on the same topic, like:

10.3390/children8090762

2) Please make sure that all studies reported in outcomes are actually studies concerning pediatric overweight/obesity.

3) The manuscript text often reports different font body, and occasionally in bold, like it were copied&pasted. Please revise carefully.

Round 2

Reviewer 1 Report

The authors present a review focused on the lifestyle and pharmacological interventions for the management of childhood obesity, which is a main and relevant topic on public health.  In this new version the authors have improved the manuscript substantially, however, there is still a issue regarding the methodology, as presented it is not clear how they performed the searching or discriminate the articles to be included, therefore the manuscript lacks of a rigorous methodology, it is not a systematic review nor a analytical review, which reflects on the variety of information (lifestyle and pharmacological) and type of included articles.

The objective was to carry out an updated review, the authors could demilit it to the use of drugs for the management of obesity, based on this premise, it is not clear what are the advantages  of this review would be with respect to integrative clinical practice guidelines that are published updated.

Reviewer 2 Report

Why methodology section was deleted?

Figure 1 is just a copy and paste from other publications. It needs to be your own work.

Please add a paragraph or two on what policy action is needed to reduce/manage obesity? What policy action need to come together?

There are a lot of abbreviations used in this review, and I think it will be more convenient for readers to read ALL. Please move the list to the end of the paper.

Round 3

Reviewer 2 Report

Dear Authors,

I still have two minor comments.

1. Is this a systematic review? I don't think so. What is the difference between a systematic review paper and a review paper? I believe this is a "literature review" or "narrative review". Thus, PRISMA diagram (Figure 2) is not needed. Please also delete "the only limitation was the language .............the electronic search".

2. Whilst the paper provides some policy recommendations, it may benefit from some broader international context to appeal to a geographically wider readership. For example it would have been useful to have drawn reference to the WHO school policy framework: http://www.who.int/dietphysicalactivity/schools/en/and perhaps some of the other policy work from Australia, Canada, the US and the UK? I would suggest referring to these interesting articles (Children (Basel). 2018 Jan 29;5(2):18; Int J Environ Res Public Health. 2021 Mar 25;18(7):3411; Int J Environ Res Public Health. 2020 Nov 13;17(22):8405; J Sport Health Sci. 2021 May;10(3):255-262; PLoS One. 2013 Oct 30;8(10):e78298).

Author Response

Response to Reviewer 2  Comments

Point 1:  Is this a systematic review? I don't think so. What is the difference between a systematic review paper and a review paper? I believe this is a "literature review" or "narrative review". Thus, PRISMA diagram (Figure 2) is not needed. Please also delete "the only limitation was the language .............the electronic search".

Response:

You are right, it is a narrative review thereby PRISMA diagram has been removed,

Point 2: Whilst the paper provides some policy recommendations, it may benefit from some broader international context to appeal to a geographically wider readership. For example it would have been useful to have drawn reference to the WHO school policy framework: http://www.who.int/dietphysicalactivity/schools/en/and perhaps some of the other policy work from Australia, Canada, the US and the UK? I would suggest referring to these interesting articles (Children (Basel). 2018 Jan 29;5(2):18; Int J Environ Res Public Health. 2021 Mar 25;18(7):3411; Int J Environ Res Public Health. 2020 Nov 13;17(22):8405; J Sport Health Sci. 2021 May;10(3):255-262; PLoS One. 2013 Oct 30;8(10):e78298).

Response:

Thank you for your remarks, I think the paper has improved significantly, so a new paragraph on the prevention strategies of childhood obesity with two new references suggested by you has been added in page 4 (Prevention section).

“Considering that children aged 6-18 years spend a large part of their day in school, it is clear that school environment is the ideal setting for childhood obesity intervention programs. The WHO school policy framework suggests the formation of wellness councils (by various school stakeholders), a published policy and programs, supporting the adoption of healthy diets and physical activity. Schools should provide students with daily physical education and should have the necessary facilities and equipment. Governments should adopt policies that support the availability of healthy diet at school and limit the availability of unhealthy products, with high salt, sugar and fat content [22].

     Nowadays in many countries, including the UK, the USA, Australia, there has been significant progress in preventing childhood obesity, by setting prevalence goals, establishing national guidelines, streamlining surveillance systems and promoting public education (through community events, mass media and social media campaigns) [23]. Participation of teachers, trained by health professionals, to promote healthy behaviors and acting as role-models, provision of additional hours to physical activity (e.g extra 30΄ of physical activity/ school day), creation of after school physical activity programs and education classes that support healthy eating habits, parental engagement on nutrition education and food preparation skills, activities such as dance classes and cooking lessons, mobilization of stakeholders to find the required funds for healthy breakfast, lunch and snack programs in schools, have been suggested as effective strategies [22,23].”

New references

  1. http://www.who.int/dietphysicalactivity/schools/en
  2. Chalkley A, Milton K. A critical review of national physical activity policies relating to children and young people in England. J Sport Health Sci. 2021 May;10(3):255-262. doi: 10.1016/j.jshs.2020.09.010.